# Lupus Nephritis Risk Factors and Biomarkers: An Update

**DOI:** 10.3390/ijms241914526

**Published:** 2023-09-25

**Authors:** Yves Renaudineau, Wesley Brooks, Julie Belliere

**Affiliations:** 1Department of Immunology, Referral Medical Biology Laboratory, University Hospital of Toulouse, Institut National de la Santé Et de la Recherche Médicale (INSERM) U1291, Centre National de la Recherche Scientifique (CNRS) U5051, 31400 Toulouse, France; 2Department of Chemistry, University of South Florida, Tampa, FL 33620, USA; wesleybrooks@usf.edu; 3Department of Nephrology and Organ Transplantation, Referral Centre for Rare Kidney Diseases, University Hospital of Toulouse, INSERM U1297, 31400 Toulouse, France; belliere.j@chu-toulouse.fr

**Keywords:** lupus nephritis, renal biopsy, urinary markers, autoantibodies, biomarkers

## Abstract

Lupus nephritis (LN) represents the most severe organ manifestation of systemic lupus erythematosus (SLE) in terms of morbidity and mortality. To reduce these risks, tremendous efforts have been made in the last decade to characterize the different steps of the disease and to develop biomarkers in order to better (i) unravel the pre-SLE stage (e.g., anti-nuclear antibodies and interferon signature); (ii) more timely initiation of therapy by improving early and accurate LN diagnosis (e.g., pathologic classification was revised); (iii) monitor disease activity and therapeutic response (e.g., recommendation to re-biopsy, new urinary biomarkers); (iv) prevent disease flares (e.g., serologic and urinary biomarkers); (v) mitigate the deterioration in the renal function; and (vi) reduce side effects with new therapeutic guidelines and novel therapies. However, progress is poor in terms of improvement with early death attributed to active SLE or infections, while later deaths are related to the chronicity of the disease and the use of toxic therapies. Consequently, an individualized treat-to-target strategy is mandatory, and for that, there is an unmet need to develop a set of accurate biomarkers to be used as the standard of care and adapted to each stage of the disease.

## 1. Introduction

Lupus nephritis (LN) is a frequent manifestation of systemic lupus erythematosus (SLE) for which current therapies are insufficiently effective and, in some cases, toxic due to important individual differences [1]. Indeed, LN is a major contributor of morbidity with about 50% of LN patients eventually developing end-stage kidney disease (ESKD), and LN is a contributor of mortality as the survival rate at 10 years is 81–98%. Therefore, in order to improve LN management, there is an unmet need to decipher the immune mechanisms to begin early monitoring of the different steps of the disease from tolerance breakdown at the pre-clinical stage, to establish LN onset as early as possible, to follow renal disease activity/flare, to optimize the therapeutic choice, to reduce toxic/infectious side effects, and to control alterations in the renal function. The pathophysiology and related biomarkers reviewed here are summarized in Figure 1.

At LN diagnosis, the gold standard technique is kidney biopsy that allows for LN histological assessment and prognosis, but its use is limited due to invasiveness. Then, to establish an active LN and monitor LN therapeutic response and relapses, the classical LN biomarkers include complement consumption with C3, C4 plus total complement hemolytic activity (CH50), and the detection of anti-double-stranded (ds)DNA antibodies (Abs) or their derivative anti-chromatin Abs, plus anti-C1q Abs and 24 h proteinuria. However, these classical LN biomarkers are limited in their capacity to predict an active LN and early relapse/remission. Then, new biomarkers have to be introduced and compared to classical LN biomarkers, in controlled and multi-centric trials, in order to be included as the standard of care. The description of classical and new LN biomarkers is the main objective of this review.

## 2. Epidemiology of LN

LN usually develops within five years following onset of the disease, and several demographic risk factors have been associated with LN development (Table 1):A young age at SLE diagnosis, which favors LN with a cut-off fixed at 33 years old [2]. This explains that LN frequency (80% vs. 30–50%) is two-fold higher in patients with juvenile-onset as compared to adult-onset together with a more severe disease and fatal evolution with 10% ESKD in juvenile-onset as compared to 4% in adult-onset [3,4].The prevalence of LN varies by race and ethnicity with African/Caribbean and Asian/Pacific islanders at high risk (35–80%), Hispanic (30–50%) at moderate risk, and Caucasian at low risk (15–25%). In addition, it has been further demonstrated that black SLE patients with African ancestors are characterized by a more aggressive renal disease when considering histological lesions, disease activity scores, lower therapeutic response, and worse outcome as compared to white patients [5].A higher frequency is reported in males as compared to females (1.1×–1.7×), which is not influenced by race/ethnicity. This contrasts with the strong sex bias reported in SLE especially during childbearing for females (female-to-male ratio: 6–13:1). SLE males have an older age at disease onset but early damage; however, conflicting results between males and females, most probably in conjunction with their racial/ethnic background, are reported when considering histological classification, disease activity, therapeutic response, flares, ESKD, and mortality [6,7].Heritability is high in SLE, which results in a relative risk to develop SLE estimated at 30% in monozygotic twins to 5–7% in dizygotic twins and first-degree relatives, with males being rarely affected [8]. Among LN patients with a familial SLE history, fever appeared to be more frequent as well as a poor renal outcome [9,10]; this is contrasted by a lower risk to develop LN, photosensitivity, and thrombocytopenia as compared to non-familial SLE [11].

**Table 1 ijms-24-14526-t001:** Demographic risk factors and lupus nephritis (LN) [2,5,9,10,11,12,13].

Epidemiological Factor	Predict LN	Disease Activity	Therapeutic Response	Flares	ESKD
Juvenile SLE	2× in juvenile	Juvenile > adult	Adult > juvenile	Juvenile > adult	Juvenile > adult
Race/ethnicity (black/white)	2× in black	Black > white	White > black	Black > white	9× in black
Male vs. female	1.1×–1.7×	Conflict results	Conflict results	Conflict results	Conflict results
Familial cases	Protective	Variable	Unknown	Unknown	High
Monogenic SLE	Yes (<5 years old)	High	Unknown	Unknown	Unknown

Abbreviations: ESKD: end-stage kidney disease; SLE: systemic lupus erythematosus; vs.: versus.

## 3. Renal Histology

Kidney biopsy is central in the diagnosis of LN and this gold standard exam is proposed in case of persistent proteinuria (>0.5 g/day) especially if associated with hematuria and/or cellular casts in the urinary sediment. The histological information obtained from kidney biopsy refers to the presence and the localization of glomerular immune complexes (IgG/A/M isotypes and complement C1q/C3/C4 deposits), and, within glomerular and tubulointerstitial components, for the appreciation of activity (six histological features) and chronicity (four histological features) [14] (Table 2). Patients with proliferative LN (class III/IV) present the worst prognosis without treatment, and the higher rate of refractory response as compared to isolated membranous LN (class V) [15]. When present, a moderate to severe interstitial fibrosis and tubular atrophy (IFTA, >25%) at baseline predicts ESKD evolution.

A second renal biopsy is highly recommended after 0.5–1 years of treatment with a persistent proteinuria (>0.7 g/day) and/or a remaining positivity to anti-dsDNA Abs in order to assess refractory LN and/or to adjust therapies [16,17]. Indeed, in those cases, a class switch from non-proliferative to proliferative LN is frequently reported in up to 50% of cases; then, the second biopsy, but not the first one, can help to adapt therapies and predict ESKD evolution [18].

On biopsy, distinguishing glomerular complement fractions via C1q/C3 staining may be informative as in a subset of LN patients, an activation of the alternative complement pathway that sustains inflammation (histology C3+/C1q− and normal serum complement C3/C4/CH50 levels) is associated with a less active disease but with a more progressive disease that affects therapeutic response and ESKD evolution [19,20]. In contrast, glomerular complement fractions via C1q (+/−C3) binding are associated with disease activity, flares, and serum complement C3/C4/CH50 consumption.

**Table 2 ijms-24-14526-t002:** Kidney biopsy and related lupus nephritis (LN) biomarkers [18,21,22,23].

	Predict Disease Activity	Therapeutic Response	Predict Flares	Predict ESKD
Proliferative LN (class III/IV ± V)	Yes (1st/2nd biopsy)	Non-responder (2nd > 1st biopsy)	No	Yes (2nd > 1st biopsy)
IFTA (Chronicity)	No	Responder (>25%)	No	Yes (baseline > 25%)
TMA	Yes (independent)	Yes (independent)	Unknown	Yes (independent)
Tissular C1q+/C3+/−	Yes	Responder	Yes	No
Tissular C1q−/C3+	No	Non-responder	No	Yes

Abbreviations: ESKD: end-stage kidney disease; IFTA: interstitial fibrosis/tubular atrophy (chronicity index); thrombotic microangiopathy; TMA: thrombotic microangiopathy (independent risk factor).

## 4. Genetic/Epigenetic/Transcriptomic Pathways and Related Biomarkers in LN

In a small subset of patients, SLE is monogenic (1–3%) with an early onset that can occur before the fifth birthday together with LN in most of the cases (80–100%) [24,25,26]. Genes implicated in monogenic cases are related to (i) DNA/RNA clearance (e.g., *DNASE1L3*, *DNASE1*, *RNASEH2C*); (ii) the complement pathway (e.g., *C1q/r/s*, *C3*); and (iii) DNA/RNA nucleic acid sensing (e.g., *TLR7*, *IRF7*) and downstream type I interferon (IFN) pathway activation (Table 3). These pathways are interconnected through the finding that cell-free RNA/DNA nucleic acids can bind autoAbs, and in turn, immune complexes are effective in inducing type I IFN production in dendritic cells and B cells. This process is amplified when upstream, the pool of cell-free RNA/DNA nucleic acids is increased (e.g., UV light, apoptosis, NETosis…), or when downstream, there is a reduction in cell-free RNA/DNA nucleic clearance. The defective clearance may result directly from a reduction in the RNA/DNA nuclease activity or indirectly from an altered capacity to opsonize cell-free DNA/RNA immune complexes in a complement-dependent process (see below).

The use of large genome-wide association studies (GWAS), epigenome-wide association studies (EWAS), and transcriptomic approaches in sporadic and polygenic SLE patients has further led to the identification of more than 100 susceptibility genes and to the characterization of additional pathways including altered programmed cell death (e.g., *FAS*, *FASL*), dysfunctional immune clearance of immune complexes (e.g., *FcGR*, *FcRn* in addition to complement fractions), the inflammatory and innate immunity phase, and the amplification of the immune response by T and B cells [27,28]. In addition, genetic variants implicated in the kidney function (e.g., *APOL1*, *MYH9*), which can impact, in turn, ESKD and mortality, have been further highlighted.

One of the most important genetic risk factors associated with SLE is related to the human leukocyte antigen (HLA) region with *HLA-DR3* and *-DR15* retrieved as risk factors for LN, whereas *HLA-DR4* and *-DR11* are protective [29]. When introduced into lupus-prone mice, the human *HLA-DR3* allele drives anti-Sm Ab production in addition to a severe glomerulonephritis and elevated anti-dsDNA Ab titers, which support a role for *HLA-DR3* in generating autoreactive T and B cells to Sm/RNP [30].

**Table 3 ijms-24-14526-t003:** Pathways and selected genetic/epigenetic/transcriptomic factors associated with lupus nephritis (LN) [31,32,33,34].

Pathways	Factors	Related Biomarkers
DNA/RNA clearance	*DNASE1L3 **, *DNAse1 **, *RNASEH2A/B/C **	DNAse activity
Complement	*C1q **, *C1s **, *C1r **, *C2 **, *C4 (CNV)*, *C3 **	C3, C4, CH50
DNA/RNA sensing & Type I IFN	*TLR7 **, *IFIH1/MDA5 **, *IRF7 **, *IRF5*, *TASL*, *IRAK1*, *STAT4*, *IFI16*	Type I IFN signature, IFN-α, anti-interferon Abs
Type II IFN	*IFN-γ*	Type II IFN signature, IFN-γ

Abbreviations: IFN: interferon; DNASE: deoxyribonuclease 1; DNASE1L3: DNASE1-like 3; RNASEH2A/B/C: ribonuclease H2 subunit A/B or C; C: complement; CNV: gene copy number variations; TLR7: toll-like receptor 7; IFIH1: interferon induced with helicase C domain 1 or MDA5; IRF5/7: IFN regulatory factor 5 or 7; TASL: TLR adaptor interacting with endolysosomal SLC15A4; IRAK1: interleukin 1 receptor-associated kinase 1; STAT4: signal transducer and activator of transcription 4; IFI16: IFN gamma-inducible protein 16; *: Associated gene with Monogenic SLE.

### 4.1. RNA/DNA Clearance by Nucleases

Monogenic loss in the function of deoxyribonuclease (DNase1, DNASE1L3) and/or of ribonucleases (RNASEH2A/B/C) that digest cell-free RNA/DNA causes SLE with anti-dsDNA/RNA Abs development and a high rate of LN [31,35]. In patients with sporadic LN, up to 50% of them present neutralizing anti-DNASE1L3 Abs that recapitulate in part the clinical presentation of monogenic DNASE1L3-deficient patients with LN [33], and this is associated with a decrease in DNAse activity in LN patients [32]. Hartl et al. have further reported that the presence of neutralizing anti-DNASE1L3 Abs in LN patients represents an independent factor from the LN histological class, and this presence is associated with disease activity and therapeutic response. The introduction of DNAse activity and anti-DNASE1L3 Abs in the list of LN biomarkers needs to be further evaluated and to be compared with existing LN biomarkers, which has not been achieved to our knowledge (Table 4).

### 4.2. Classical Complement Pathway

Monogenetic complement deficiencies (C1q but also C1r, C1s, C2, and C3) or copy number variations (CNVs) in C4 alter the initiation of the classical complement pathway and, in turn, are causal for SLE/LN development [24]. Moreover, an acquired deficiency of C1q by autoAbs can be present among SLE patients, which points according to its localization (kidney and/or skin) to a proliferative LN and/or an hypocomplementemic urticarial vasculitis or MacDuffie syndrome (HUVS). Anti-C1q Abs positivity is a useful biomarker for an active and proliferative LN (60–89% in active LN vs. 0–15% in inactive LN and non-renal SLE; area under the curve [AUC] = 0.79), but anti-C1q Abs capacity to predict therapeutic response and early relapse is limited [36,37].

Altogether, this reveals the two faces of the complement system in LN: on the one hand, a functional classical complement system is critical for a proper RNA/DNA immune complex clearance, and to avoid tolerance breakdown to self. On the other hand, the deposition of glomerular (or skin, in case of HUVS) immune complexes, that mostly contain cell-free RNA/DNA nucleic acids, induces tissular injury through the recruitment of C1q via two mechanisms. Firstly, when present on immune complexes, C1q directly activates the classical complement cascade, leading to the release of the anaphylatoxins and to the formation of the lytic membrane attack complex. This complement activation can be monitored by exploring in situ glomerular (or skin) C1q deposits or ex vivo by testing serum C3/C4/CH50 consumption. Secondly, and as demonstrated by Trouw et al., the presence of C1q on glomerular immune complexes can participate in the recruitment of anti-C1q Abs and then to the recruitment of activated polynuclear cells to form a super cellular immune complex [38]. In other words, and to exert their pathogenic effect, anti-C1q Abs necessitate the presence of an immune complex deposit in the kidney (or skin), and this process predominates in proliferative LN (or HUVS).

### 4.3. Type I and Type II Interferons

Renal dysfunction is one of the hallmarks of monogenic diseases characterized by a constitutive activation of the type I IFN pathway, referred to as interferonopathy, and part of them overlap with SLE [39]. In addition to genetic studies, high-throughput immune monitoring of polygenic SLE patients has further confirmed the expression of IFN-stimulated genes (ISGs) together with epigenetic markers at ISGs in SLE patients [40,41,42]. The reliable strategy to explore the IFN pathways remains to test a set of ISGs, referred to as an IFN signature, as the measurement of serum IFN-α is challenging due to the poor sensitivity and specificity of the immunoassay routinely used, thus allowing the estimation of cell exposure to the three types of IFN [43,44]. Type I, and to a lesser extent type III IFN, are produced by nucleated cells to elicit an inflammatory and anti-viral response, while type II IFN is primarily expressed by lymphocytes and more specifically by CD4+ T helper 1 cells, CD8+ cytotoxic T cells, and NK cells to regulate the immune response.

At the pre-clinical stage during tolerance breakdown and years before SLE onset, the detection of anti-dsDNA Abs is concomitant with a type II IFN signature, while a type I IFN signature starts to be detected later at SLE onset [45]. SLE patients with a positive type I IFN signature have increased renal involvement and elevated anti-dsDNA Ab levels [46], and when both type I and II IFN signatures are present, they are associated with disease activity, flares, and disease injury in the kidney, skin, and nervous system [44,47].

To go further in the analysis of the IFN signatures in the kidney, a cellular IFN landscape at the single-cell level and renal microdissection have been conducted in LN patients to establish that [27,48,49,50,51]:Proliferative class III/IV LN are characterized by prominent type I and type II IFN signatures in renal epithelial cells.Membranous class V LN can be distinguished by elevated type I IFN and tumor necrosis (TNF)-α signatures together with an altered cell metabolism signature in renal epithelial cells, thus suggesting distinct pathophysiology processes between proliferative and membranous LN.Type I IFN level in renal tubular cells correlates with an elevated IFTA chronicity index, which supports a role for type I IFN, reinforced or not with type II IFN, to promote ESKD.When detected in renal leukocytes, the type I IFN response drives an extrafollicular B cell response with aged/autoreactive B cells (ABC) and T follicular regulatory CD4+ T cells.An IFN signature in renal tubular cells recapitulates the IFN signature in skin biopsies, and the IFN signature in infiltrating renal leukocytes can be appreciated by exploring peripheral blood mononuclear cells (PBMCs).

To counterbalance the deleterious effect of constitutive type I IFN activation, a minor subset of SLE patients (5%) present neutralizing type I anti-IFN Abs and this subset is associated with lower disease activity [52,53]. Non-neutralizing type I/II/III anti-IFN Abs can also be present in serum and urine from SLE patients (50–60%), and among them, the detection of type II anti-IFN Abs is associated with elevated anti-dsDNA Ab levels, C3/C4 complement consumption, and SLE disease activity [54,55].

Overall, the use of peripheral blood IFN signatures as biomarkers is restricted to the diagnosis of interferonopathies and to clinical trials as IFN signatures did not recapitulate all SLE patients and a bias exists according to the ancestry [56,57].

## 5. Nephropathic Antibodies in LN

In contrast to other autoimmune kidney diseases that are restricted to a specific autoAb (e.g., anti-collagen IV Abs in glomerular basement membrane), a vast array of autoAbs characterize LN patients with distinct subsets present in the kidney [58]. Indeed, the pioneering experiment conducted by Mannik et al. from 23 post-mortem kidneys revealed the presence of a large panel of polyreactive Abs targeting dsDNA/chromatin/histone, Sm/RNP, SSA/B, C1q, and phospholipids [59]. Thanks to lupus-prone animal models and high-throughput technologies, the list of LN-associated Abs has been extended (Table 5), and this list included:Abs targeting RNA/DNA nucleic acids.Abs targeting glomerular antigens, particularity the part of them which cross-react with anti-dsDNA Abs.Functional Abs targeting pathophysiological pathways (e.g., anti-DNAse1L3 Abs, anti-C1q Abs, anti-IFN Abs, please refer to Section 4).Abs targeting anti-phospholipids and cofactors.

**Table 5 ijms-24-14526-t005:** Lupus nephritis (LN)-associated autoantibodies (see also Table 4) [60,61].

Autoantibody (Ab)	Predict LN (Histology)	Predict Disease Activity	Therapeutic Response	Predict Flares	Predict ESKD
Anti-dsDNA Abs	High levels (III & IV > V)	High levels	Responder	High levels	Low
Anti-Sm Abs	No	Suspected	Unknown	Suspected	Suspected
Anti-SSB Abs	No	No	No	No	No
Anti-α actinin Abs	Yes	Yes	Unknown	Unknown	Unknown
Anti-CL/β2 GPI Abs	No	No	Low	No	Yes (thrombotic microangiopathy)

Abbreviations: ESKD: end-stage kidney disease; dsDNA: double-stranded DNA; Sm: Smith; SSB: sicca syndrome B; CL: cardiolipin; β2 GPI: beta 2 glycoprotein I.

### 5.1. Abs Targeting Nucleic Acids

Among antinuclear Abs, a dichotomy exists between anti-dsDNA and anti-ribonucleoprotein (RNP) Abs that covers anti-Sm/RNP-A/RNP-U1 and anti-SSA/B Abs. Indeed, these two groups are suspected to result from distinct B cell responses: namely, the extracellular pathway with short-lived plasma cells for the former and the germinal center pathway with long-lived plasma cells for the latter [62,63]. As a consequence, anti-nucleic acid Abs are serially detected up to 5 years before onset at the non-specific pre-clinical stage for anti-SSA/SSB Abs, at 2 years before SLE onset for anti-dsDNA Abs, and when present at SLE onset, for anti-Sm Abs [64]. These differences between anti-dsDNA and anti-RNP Abs explain that therapeutic monoclonal (m)Abs targeting type I IFN (anifrolumab) or downstream the ISG BAFF (belimumab) are effective in controlling anti-dsDNA Abs but not anti-RNP Abs, while therapeutic strategies targeting tissular B cells and long-/short-lived plasma cells can lead to both RNA/DNA sero-negativity (to be discussed in more detail below in the therapeutic part) [65,66,67].

Based on their performance to discriminate SLE from other autoimmune diseases, both anti-dsDNA Abs and anti-Sm Abs have been included in SLE classification criteria [68]. Nevertheless, the spectrum of anti-dsDNA/chromatin Abs is heterogeneous with various chromatin structures recognized (nucleotides, sugar-phosphate backbone, helicoidal Z-form, 3D chromatin, and histones), DNA:RNA hybrids, DNA:peptide hybrids, and modified nucleic acids among other structures [69]. As a consequence, at SLE/LN diagnosis, it is recommended to use different techniques such as an enzyme-linked immunosorbent assay (ELISA) and indirect immunofluorescence (IFI) on *Crithidia luciliae* for anti-dsDNA Abs, which can be improved by supplementing with anti-chromatin Abs [70,71]. Another recommendation, in the monitoring of SLE/LN patients, is to use the technique that best predicts disease activity and flares from the same laboratory.

Anti-dsDNA Abs tested by ELISA are classically used to monitor disease activity in LN, explaining that anti-dsDNA Ab positivity accounts for two points in the SLE disease activity score (SLEDAI)-2K [72], although the performance is weak in discriminating active LN (AUC = 0.6–0.7) [73]. The presence of anti-dsDNA/chromatin Abs in the follow-up of LN patients represents a risk factor for relapse but its sensitivity and specificity are weak in predicting early relapse, and levels decrease following the introduction of therapy [74,75]. Although not useful for predicting the LN histology class among active patients, anti-dsDNA Ab levels are higher in proliferative LN (class IV/IV) as compared to membranous LN (isolated class V). SLE patients with anti-dsDNA Abs but negative for anti-RNPs Abs presented a mild type I IFN response in association with complement consumption, while isolated anti-RNP Abs are characterized by a strong type I IFN response in the absence of complement consumption [62]. The utility of anti-SSA/SSB and anti-Sm/RNP Abs in the follow-up of LN patients is controversial, among them, anti-Sm Abs, which presented an important racial/ethnic bias (30% in black vs. 5% in white) and an overlap with anti-dsDNA Abs, and are described to be imperfectly associated with milder/high renal involvement, renal flares, and Raynaud’s phenomenon [76,77,78].

### 5.2. Anti-dsDNA/Glomerular Antibodies

LN-pathogenic anti-dsDNA monoclonal (m)Abs have been defined through their capacity to induce LN following their introduction into animal models [79,80]. One of the characteristics of these LN-pathogenic mAbs (e.g., clone R4A) is to bind non-DNA structures in renal tissues treated with DNases and related targets including α-actinin, laminin, annexin II, ribosomal P, collagen III/IV, entactin, proteoglycan heparan sulfate, and N-methyl-D-aspartate receptors (NMDAR) [81,82]. In humans, these cross-reactive anti-dsDNA/α actinin-glomerular Abs have been retrieved at LN onset with the particularity to be effective in binding kidney cells including mesangial cells, podocytes, and the renal cell line HEK [83,84,85]. Glomerular Abs have not demonstrated their utility in the follow-up of LN patients as biomarkers [86,87,88].

### 5.3. Anti-Phospholipid Antibodies

A persistent detection of anti-phospholipids (>12 weeks) is retrieved in 20–40% of SLE patients, which increases the risk of arterial/venous thrombosis and/or pregnancy morbidity [89]. Among LN patients, 5–15% of them present glomerular microvascular damage at biopsy that defines anti-phospholipid nephropathy (APSN) [90], and this distinct histological feature and independent risk factor for ESKD development results from the presence of an acute and/or chronic thrombotic microangiopathy (TMA) together with serum anti-phospholipid Ab detection [91,92]. APSN is multi-factorial and attributed, in addition to anti-phospholipid Abs, to complement system activation with C4d staining, inflammatory factors, infections, genetic factors (e.g., HLA-DRB1*13), and drugs [93]. At LN onset and to lower the risk of hemorrhage, anti-phospholipid Abs have to be explored and, in case of positivity, an oral anti-coagulation can be considered although no standard recommendation exists.

## 6. Urinary Biomarkers Associated with LN

Glomerular deposition of extra-renal immune complexes containing RNA/DNA nucleic acids and activation of the complement system was initially proposed as the main mechanism leading to LN [94,95]. In this process, activation of the classical complement pathway leads to the release of anaphylatoxin (C3a, C5a) that act as chemoattractants to innate and acquired immune cells causing inflammatory mediator release, such as type I IFN, which amplify glomerular lesions. In proliferative LN (class III/IV), mesangial cells are targeted via subendothelial immune complex deposition and complement activation, leading to their proliferation, extracellular matrix production, and an important pro-inflammatory cytokine production including type I IFN plus IL1-β, IL6, and IL8. In membranous LN (class V), differences are related to the immune complex localization that is subepithelial, to limited complement and inflammatory reactions, and to podocyte dysfunctions associated with severe proteinuria [96]. Resident lymphocytes are also important in LN development, which has been attributed to the characterization in the kidneys from some LN patients of T and B cells forming extrafollicular germinal centers that maintain a local production of autoAb, CD8+ cytotoxic T cells, and pro-inflammatory cytokines including type II IFN [97]. Last but not least, renal cytokine and complement C3a anaphylatoxin release can promote macrophage polarization into M2 phenotype (CD163+), a dominant pro-inflammatory and pro-fibrotic population present in acute glomerular and tubulointerstitial lesions [98,99]. The number of glomerular M2 (CD163+) cells are correlated with disease activity, proteinuria, and cellular crescent that may evolve to fibrosis and ESKD [100,101]. CD163 acts as a scavenger receptor for hemoglobin and heme, and can undergo shedding via an inflammatory stimulus, which constitutes a stable urinary biomarker to estimate the glomerular M2 population.

Advances in understanding the pathophysiology of LN have further contributed to expanding the list of LN-associated urinary biomarkers to (i) kidney markers (proteinuria); (ii) pro-inflammatory cytokines/chemokines; and (iii) cell adhesion molecules. These biomarkers that are non-specific for SLE can be classified according to their capacity to predict an active LN/glomerulonephritis, therapeutic response, flares, and ESKD, as described in Table 6.

**Table 6 ijms-24-14526-t006:** Lupus nephritis (LN)-associated urinary biomarkers [13,102,103,104].

Biomarker	Active LN/Glomerulonephritis	Therapeutic Response	Predict Flares	Predict ESKD
Renal markers	24 h proteinuria, SUA, uGAL3BP	24 h proteinuria	24 h proteinuria	24 h proteinuria, SUA, GFR
Cytokine/chemokines	TWEAK, MCP-1/CCL2	TWEAK, MCP-1/CCL2, BAFF	MCP-1/CCL2	TWEAK, MCP-1/CCL2
Cell adhesion molecules	ALCAM, VCAM, NGAL, KIM1, sCD163	VCAM1, sCD163,	VCAM1, sCD163, KIM1	NGAL, ALCAM, VCAM1, KIM1, sCD163
miRs	miR-146a, miR-204, miR-30c, miR-3201, miR-1273e	miR-135	miR-146a	miR-146a

Abbreviations: ESKD: end-stage kidney disease; SUA: serum uric acid; uGAL3BP: urinary galectin-3 binding protein; GFR: glomerular filtration rate; TWEAK: tumor necrosis factor (TNF)-like weak inducer of apoptosis; MCP-1/CCL2: monocyte chemoattractant protein-1 or CCL2; BAFF: B-cell-activating factor of the tumor necrosis factor family or BLySS; VCAM1: vascular cell adhesion molecule; ALCAM: activated leukocyte CAM; NGAL: neutrophil gelatinase-associated lipocalin; KIM1: kidney injury molecule-1; miR: micro-RNA.

### 6.1. Proteinuria and Protein to Creatinuria Ratio

The gold standard for proteinuria assessment is the 24 h urine collection assay, but this is a cumbersome and often impractical approach so the proteinuria to creatinuria ratio (PCR) is used as a surrogate, based on a good to high degree of correlation between the two parameters [105,106]. An elevated proteinuria (>0.5 g/24 h) is used as the criteria to perform a renal biopsy and confirm LN in SLE patients [14], and proteinuria (>0.5 g/24 h) with/without hematuria (>5 red blood cells/high-power field) is used to define SLE disease activity (4 + 4 points in SLEDAI-2K). Although there is no consensus to define a treatment response in LN, the Kidney Disease Improving Global Outcomes (KDIGO) clinical practice guideline considers complete remission, within 6–12 months of initiating therapy, when the PCR is <0.5 g/g with a stabilized estimated glomerular filtration rate (eGFR) from the baseline [107]. According to the recommendations of the Joint European League Against Rheumatism and European Renal Association-European Dialysis and Transplant Association (EULAR/ERA-EDTA), a complete remission can be considered, after 12 months of therapy, when both the 24 h proteinuria is <0.5–0.7 g and the eGFR is (near)normalized [17]. Importantly, renal damage (e.g., ESKD, nephrotic syndrome) and urinary tract infections are the main limitations to the use of proteinuria to monitor LN patients, as these two conditions can affect the capacity to indicate LN activity/flare and therapeutic response evaluation.

When used to monitor LN patients under treatment, an elevated proteinuria at baseline (>1–2 g/24 h) negatively predicts complete remission [18], and a partial improvement or an increase in proteinuria (>2 g/24-h) are likely to predict relapse and progression to ESKD [108,109]. In addition to the proteinuria, additional kidney markers can be used. One example is the serum hyperuricemia (>5–7 mg/dL) that can be used to predict LN, disease activity, and ESK progression [110,111]. Another example is the urinary galectin-3 binding protein (uGAL3BP) that can also help to discriminate LN patients from non-renal SLE patients and to predict disease activity [112,113].

### 6.2. Cytokines/Chemokines

The top five urinary cytokines/chemokines evaluated in LN included: tumor necrosis factor (TNF)-like weak inducer of apoptosis (TWEAK); monocyte chemoattractant protein-1 (MCP-1/CCL2); IFN inducible protein-10 (IP-10/CXCL10); B-cell-activating factor of the tumor necrosis factor family (BAFF/BLySS); and proliferation-inducing ligand (APRIL) [114,115]. Among them, biomarkers with a good/very good AUC for predicting LN included urinary MCP-1/CCL2 (AUC = 0.90) > BAFF (AUC = 0.825) = TWEAK (AUC = 0.82) > IP-10 (0.6 < AUC < 1.0) [116,117]. When combined, urinary MCP-1/CCL2 and TWEAK are effective in discriminating between an active and inactive LN (AUC = 0.89), and ESKD evolution (AUC = 0.78). Variations in urinary MCP-1/CCL2 levels can be used to monitor complete response after initiating therapy and flares [118,119].

### 6.3. Cell Adhesion Molecules

Urinary cell membrane biomarkers useful in LN include the cell adhesion molecules (CAM) with vascular CAM1 (VCAM1) and activated leukocyte CAM (ALCAM), the neutrophil gelatinase-associated lipocalin (NGAL), the kidney injury molecule-1 (KIM-1), and the soluble (s)CD163 receptor shed from pro-inflammatory/fibrotic macrophages M2. In order to improve their performance, urinary levels can be normalized as the ratio with creatinuria, or less frequently with proteinuria or its serum fraction [120,121]. A combination of urinary biomarkers can be further used to discriminate proliferative LN from membranous LN [37].

Although not specific for SLE, an elevated level of urinary cell adhesion molecules exhibited excellent accuracy for distinguishing active from inactive LN and from non-renal SLE with a very good AUC > 0.8, except for NGAL [122,123,124]. All these urinary cell membrane biomarkers can be used to monitor the treatment response and flares [125], and at baseline, the NGAL levels best predict, as compared to VCAM and KIM1, responders from non-responders 6 months after the induction phase with an AUC = 0.78 [37]. ROC and/or Cox regression analysis further showed that persistently high urinary VCAM1, NGAL, KIM1, and sCD163 levels can predict ESKD evolution [37,116,125,126]. The added value to combine these markers needs to be further evaluated.

### 6.4. mi-RNAs

Among epigenetic alterations reported in SLE, a list of 5 urinary miRNAs (miR-146a, miR-204, miR-30c, miR-3201, miR-1273e) present in extracellular vesicles have been associated with LN in a meta-analysis from 11 LN studies [127]. These miRNAs are associated with LN-related pathways (e.g., RNA/DNA nucleic acid and inflammatory pathways) and renal homeostasis (e.g., WNT and TGF-beta signaling). An association between renal flare and ESKD is reported with miR-146a levels at baseline [128]. The urinary miRNA-135b produced by tubular cells is differentially expressed between responders and non-responders [129]. Urinary miRNA detection remains restricted to clinical trials and research laboratories.

## 7. Lessons from Therapeutic Trials

LN therapy is based on two pillars: firstly, an induction phase (up to 6 months) with the objective to achieve complete renal remission followed by a second phase of maintenance to limit relapses, avoid ESKD, and reduce systemic side effects. During these two phases, the standard of care (SOC) includes the use of glucocorticoids and antimetabolite/immunosuppressive drugs (e.g., mycophenolate mofetil or cyclophosphamide). After the maintenance phase, and in order to improve complete remission, add-on therapies can be proposed in addition to the SOC. To this end, the European Medicines Agency (EMA) allows three add-on therapies targeting the type I IFN receptor (anifrolumab), the ISG BAFF/BLyS (belimumab), or the non-nephrotic calcineurin inhibitor voclosporin. Ongoing trials further support in refractory SLE/LN to target the type I/I IFN JAK/STAT pathway (e.g., tofacitinib), to deplete tissular B cells ± plasmocytes (CAR T cells and obinituzimab instead of rituximab), or to deplete CD38-positive plasmocytes (daratumab) [130].

The use of antimetabolite/immunosuppressive drugs and glucocorticoids as the SOC is associated with remission in 20–40% of cases that can be monitored by exploring complement fractions, anti-dsDNA Abs levels, anti-C1q Abs levels, and in research trials, type I IFN signature (Table 7). Such an effect can be improved through the use of add-on therapies as compared to the SOC:A better control of the type I IFN pathway and anti-dsDNA Abs levels as reported with Jak inhibitors, and control of anti-dsDNA/C1q Abs levels with voclosporin, while complement fraction levels remain unaffected with these two drugs [131,132].A better control of the type I IFN pathway, complement fractions, and anti-dsDNA/C1q Abs levels is reported with anifrolumab and belimumab [133,134]. Post-hoc analysis has further reported that responders presented an elevated type I IFN signature at baseline, which supports using this biomarker for therapeutic decisions although it has not been validated in a controlled trial [135,136]. According to Weeding et al., a reduction in urinary sCD163 best predicts a complete response when belimumab is used as an add-on therapy [137].A better control of complement fractions and anti-dsDNA/C1q Abs levels is reported with all anti-CD20 mAbs, but among them, only obinutizumab (class II anti-CD20 mAbs) was effective in improving renal response compared with the SOC [66,138,139,140]. Future directions regarding anti-B cell therapies in LN are related to the use of combinations between anti-CD20 and anti-BAFF, the use of anti-CD38 mAb (daratumab) to target long-lived plasma-cells not covered by anti-CD20 mAbs, and the use of CAR T cells that have reported a complete response in six refractory SLE patients, all with LN [65,141,142,143].

**Table 7 ijms-24-14526-t007:** Lupus nephritis (LN) biomarker evolution in clinical trials.

Drugs	Renal Response vs. Controls (%)	Complement Consumption	Anti-dsDNA Abs	Anti-C1q Abs	Anti-SSA/B and Anti-Sm/RNP	Other Biomarkers
SOC (e.g., mycophenolic acid) [144,145,146]	20–40%	Improvement	Decrease	Decrease	No effect	Control Type I IFN
Jak inhibitors vs. SOC [131,147]	Similar to SOC	No effect	Decrease	Unknown	Decrease at high level	Control Type I/II IFN
Voclosporin vs. SOC [132]	44% vs. 23%	No effect	Decrease	Decrease	No effect	Unknown
Anifrolumab vs. SOC [133,135]	45% vs. 31%	Improvement	Decrease	Decrease	No effect	IFN-I high
Belimumab vs. SOC [134,136,137,148,149]	43% vs. 32%	Improvement	Decrease	Unknown	Decrease at high level	IFN-I high, sCD163
Obinituzimab (type II anti-CD20) [138]	35% vs. 23%	Improvement	Decrease	Decrease	No effect	Unknown
Refractory SLE (mostly LN)						
Belimumab + rituximab [141,150,151]	52% vs. 41%	Improvement	CR: delayed negativity	CR: decrease	CR: mild decrease	Unknown
CAR T cells [65]	100%	Normalization	Negativation	Negativation	Negativation	Unknown
Daratumab (CD38) [67,142]		Normalization	Decrease	Decrease	Unknown	Control Type I IFN

## 8. Future Challenges

As described in this review and summarized in Figure 2, various conditions can lead to LN. However, following classic biomarkers in patients, including anti-dsDNA/C1q Abs, complement fractions, and 24 h proteinuria, is limited in terms of sensitivity and specificity. Subsequently, new biomarkers have emerged from clinical trials, such as IFN signature and urinary biomarkers, to better predict LN evolution and treatment response at the patient level. However, major challenges remain with regard to biomarkers in LN.

To this end, we have identified eight future challenges regarding biomarkers in LN:Challenge 1: Better understand the molecular pathways implicated in LN and for that, single-cell multi-omic analysis may be informative (e.g., Sc-RNASeq, Sc-proteomic…) and contribute to the identification of new biomarkers in LN. Recently, kidney samples from patients with lupus nephritis and from healthy control subjects were analyzed thanks to single-cell RNA sequencing [27]. The analysis revealed 21 subsets of leukocytes active in the disease, including multiple populations of myeloid cells. Classification and annotation of myeloid cell clusters (C) revealed resident and infiltrating populations. Focused analysis of the 466 cells in myeloid clusters C4 and C6 revealed 5 finer clusters. Two of them were found to express CD163 transcripts, suggesting that such an approach can allow for the identification of the glomerular urinary sCD163-producing cells. Because these cells have been implicated in renal prognosis, it could represent a new prognostic factor to correlate to renal outcome. In-depth characterization of renal immune cells is a pre-requisite to then screen for therapeutic candidates including drug repurposing approaches using virtual and experimental screening techniques.Challenge 2: Properly evaluate biomarkers and for that, validation of such requires multi-ethnic and larger cohorts as well as well-annotated clinical cohorts with the main objectives being to provide more reliable biomarkers with a high performance in terms of sensitivity and specificity (≥90%) that can (i) replace initial/follow-up renal biopsies; (ii) determine the therapy that is the best fit for the patient; (iii) prevent ESKD/fatal evolutions; and (iv) reduce side effects.Challenge 3: Evaluate the complementarity and determine the minimal number of biomarkers combined with clinical signs to help in better guiding clinicians and researchers in their choices. For that, one example would be to develop “LN scores” integrating clinical signs with circulating and urinary biomarkers by using advances in bioinformatics using artificial intelligence, available databases from real-life cohorts, and studies on longitudinal evolution [152].Challenge 4: Propose new criteria for the evaluation of LN activity, LN fibrosis, therapeutic response/choice, and for that, more accurate biomarkers have to be selected in order to capture moderate and mild changes, and to avoid hospitalizations and invasive exams. One example is that accurate biomarkers could help both for the early detection of refractory forms of LN before the usual clinical visit, and to distinguish residual proteinuria from active proteinuria, which is currently impossible without biopsy. Another example is that nephrologists have to weigh the benefit/risk ratio for empiric steroid therapy when anti-phospholipid syndrome is associated with SLE and when patients should not interrupt anti-coagulant therapies. This is of particular complexity when LN is active and the patient has already experienced severe thrombotic complications. In such a situation, a non-invasive signature of renal activity could allow for tailoring appropriate immunosuppressive intensity to the clinical situation.Challenge 5: Characterize and recruit for future clinical trials cohorts of homogeneous LN patients with similar biological profiles (aka, endotypes), organ involvement, comorbidities, and severity.Challenge 6: Use adapted biomarkers to identify LN patients at particular risk of associated severe organ involvement and damage, for severe drug side effects, and for infection/tumor susceptibility.Challenge 7: Develop biomarkers to improve pregnancy outcomes as this is a major concern in women with SLE regarding LN flares, pre-eclampsia, and congenital heart block in the fetus involving anti-SSA Abs.Challenge 8: Propose biomarkers toward LN as health-related quality of life including, for example, fatigue, pain, and psychological behaviors which are aspects of LN of high concern for patients and result, in part, from immunological processes.

## 9. Conclusions

The biomarkers used for the management and prognosis of LN are evolving and there is no doubt that future advances in the establishment of biomarkers in LN will help to better characterize the pathophysiological pathways, allowing for improvements in the classification, follow-up, well-being, and treatment of patients with LN.

## Figures and Tables

**Figure 1 ijms-24-14526-f001:**
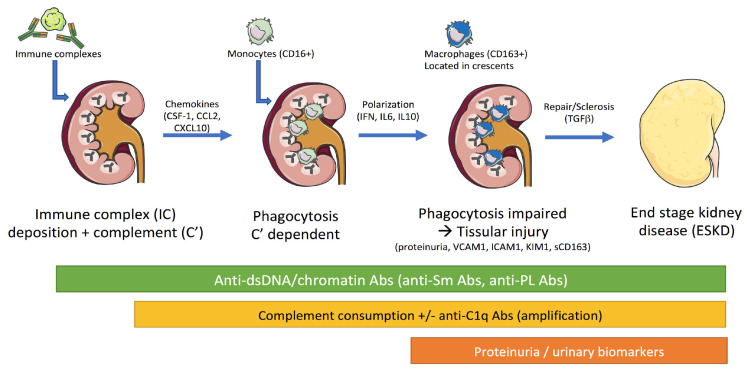
Pathogenic mechanisms and related biomarkers in lupus nephritis (LN). Accumulation of immune complexes (IC) containing nucleic acids is reported at pre-LN stage, and this process is driven by the presence of anti-nucleic acid antibodies (Abs). Next, the half-life of the renal nucleic acid immune complexes is abnormally increased that results, in part, to the polarization of macrophages implicated in renal repair and sclerosis. Activation of the classical complement is also important for tissular injury through intrarenal complement deposition, and when present, anti-C1q Abs amplify this process. This figure was produced using images from Servier Medical Art.

**Figure 2 ijms-24-14526-f002:**
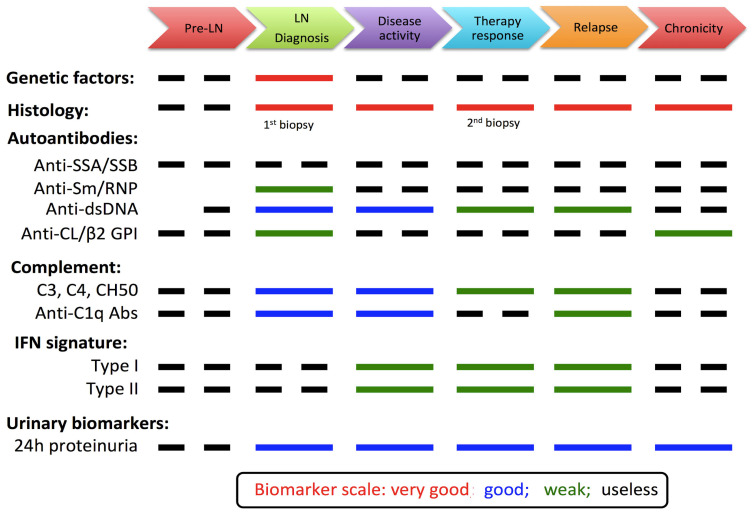
Lupus nephritis (LN) biomarkers and their level of performance used to monitor the different steps of the disease; dotted lines represent negative associations. Adapted from [37,116].

**Table 4 ijms-24-14526-t004:** Lupus nephritis (LN) signaling pathways and related biomarkers.

Biomarker	Predict LN	Disease Activity	Therapeutic Response	Predict Flares	Predict ESKD
Anti-DNASE1L3 Abs, DNase activity	Yes	Yes	Responder	Unknown	Unknown
Anti-C1q Abs	Proliferative LN	Low	No	Mild	Unknown
Serum C3/C4/CH50	No	Consumption	Responder	Consumption	No
IFN signature (I ± II); IFN-alpha	Elevated levels	Elevated levels	Responder	Elevated levels	Elevated levels

Abbreviations: ESKD: end-stage kidney disease; Abs: autoantibodies; DNASE1L3: deoxyribonuclease 1-like 3; IFN: interferon; LN: lupus nephritis.

## Data Availability

Not applicable.

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
