# Peer review of "Lupus Nephritis Risk Factors and Biomarkers: An Update"

_ijms, 2023, doi:10.3390/ijms241914526_

Round 1

Reviewer 1 Report

The paper under review addresses the crucial and challenging issue of biomarker selection for the treatment of lupus nephritis, a topic of paramount importance in the realm of autoimmune diseases. While the study presents notable general and divulgation characteristics, it calls for a more comprehensive appraisal of the primary aspect, specifically, the discussion of the biomarkers for lupus nephritis treatment.

The paper is engaging, employing a narrative style that prompts readers to ponder the significance of biomarkers in advancing lupus nephritis treatment and management. A deeper discussion of the advantages, limitations, and potential complementarity of various biomarkers would aid in better-guiding clinicians and researchers in their choices. While the paper touches upon the clinical implications of the chosen biomarkers, a deeper exploration of how these biomarkers may influence the treatment and management of lupus nephritis in practice is warranted. This could include insights into early diagnosis, tailored treatment approaches, and patient outcomes.

The paper lacks a robust discussion of future directions in biomarker research for lupus nephritis. Offering insights into emerging technologies, potential therapeutic breakthroughs, or areas of further investigation would add value to the study and inspire future research endeavors.

The paper shows a good level of clarity and precision in its writing, with only minor typos and errors.

Author Response

The paper lacks a robust discussion of future directions in biomarker research for lupus nephritis. Offering insights into emerging technologies, potential therapeutic breakthroughs, or areas of further investigation would add value to the study and inspire future research endeavors.

Answer 1: In order to answer your request a new section presenting eight future challenges regarding biomarkers in LN is now added.

A deeper discussion of the advantages, limitations, and potential complementarity of various biomarkers would aid in better-guiding clinicians and researchers in their choices. While the paper touches upon the clinical implications of the chosen biomarkers, a deeper exploration of how these biomarkers may influence the treatment and management of lupus nephritis in practice is warranted. This could include insights into early diagnosis, tailored treatment approaches, and patient outcomes. While the study presents notable general and divulgation characteristics, it calls for a more comprehensive appraisal of the primary aspect, specifically, the discussion of the biomarkers for lupus nephritis treatment.

Answer 2 : these specific points have been adresses in « challenge » number 3 and 4

Challenge 3: Evaluate the complementarity and the minimal number of biomarkers combined with clinical signs to help in better-guiding clinicians and researchers in their choices. For that, one example would be to develop “LN scores” integrating clinical signs with circulating and urinary biomarkers by using advances in bioinformatics using artificial intelligence, available databases from real-life cohorts, and studies on longitudinal evolution [153].

Challenge 4: Propose new criteria for the evaluation of LN activity, LN fibrosis, therapeutic response/choice, and for that the more accurate biomarkers have to be selected in order to capture moderate and mild changes, and to avoid hospitalizations and invasive exams. One example, accurate biomarkers could help both for early detection of refractory forms of LN before the usual clinical visit, and to distinguish residual proteinuria from active proteinuria, which is currently impossible without biopsy. Another example, nephrologists have to weigh the benefit/risk ratio for empiric steroid therapy when anti-phospholipid syndrome is associated with SLE and when patients should not interrupt anti-coagulant therapies. This is of particular complexity when LN is active and the patient has already experienced severe thrombotic complications. In such a situation, a non-invasive signature of renal activity could allow tailoring appropriate immunosuppressive intensity to the clinical situation.

Reviewer 2 Report

Renaudineau et al. report in their review the need to identify specific lupus nephritis biomarkers for diagnosis, to monitor disease activity and progression and the therapeutic response. They summarize the current knowledge and report about problems in the field.   The review is well organized. The  introduction provides a good overview about lupus nephritis and the currently used diagnostic methods and disease markers. This is important for readers not familiar within this field. The authors focus in their manuscript primarily on  “classical” disease marker such as autoantibodies, complement components  and interferon signatures and their importance in the biomarker field as well as their benefits and limitations.  They also strengths the importance of urinary biomarkers associated with lupus nephritis although some of them are not SLE specific. They emphasize the need of evaluation of biomarker utility in large clinically trials.  Very interesting and important is the comparison between standard of care treatment and new drugs approved for SLE or lupus nephritis treatment recently. . Nevertheless, I miss an outlook that in the future new lupus nephritis-specific biomarkers could be identified by RNA-Seq analyses or -omic approaches (proteomic, metabolomics). Perhaps in the future it will even be possible to identify markers that distinguish between different classes of LN and help to determine the response to treatment. A brief overview of current research in this field would complete the review.

Author Response

Nevertheless, I miss an outlook that in the future new lupus nephritis-specific biomarkers could be identified by RNA-Seq analyses or -omic approaches (proteomic, metabolomics). Perhaps in the future it will even be possible to identify markers that distinguish between different classes of LN and help to determine the response to treatment. A brief overview of current research in this field would complete the review.

Answer 1: In order to answer your request and reviewer 1 request, a new section presenting eight future challenges regarding biomarkers in LN is now added and your specific point has been addresses in « challenge » number 1.

Challenge 1: Better understand the molecular pathways implicated in LN and for that single cell multiomic analysis may be informative (e.g., Sc-RNASeq, Sc-proteomic…) and contribute to the identification of new biomarkers in LN. Recently, kidney samples from patients with lupus nephritis and from healthy control subjects were analysed thanks to single-cell RNA sequencing [27]. The analysis revealed 21 subsets of leukocytes active in the disease, including multiple populations of myeloid cells. Classification and annotation of myeloid cell clusters (C) revealed resident and infiltrating populations. Focused analysis of the 466 cells in myeloid clusters C4 and C6 revealed 5 finer clusters. Two of them were found to express CD163 transcripts, suggesting that such an approach can allow the identification of the glomerular urinary sCD163 producing cells. Because these cells have been implicated in renal prognosis, it could represent a new prognostic factor to correlate to renal outcome. in-depth characterization of renal immune cells is a prerequisite to then screen for therapeutic candidates including drug repurposing approaches using virtual and experimental screening techniques.